# Molecular mechanism for rapid autoxidation in α-pinene ozonolysis

Siddharth Iyer [1,7✉], Matti P. Rissanen [1,7✉], Rashid Valiev[2,3], Shawon Barua [1], Jordan E. Krechmer [4], Joel Thornton[5], Mikael Ehn [6] & Theo Kurtén [2✉]

Aerosol affects Earth's climate and the health of its inhabitants. A major contributor to aerosol formation is the oxidation of volatile organic compounds. Monoterpenes are an important class of volatile organic compounds, and recent research demonstrate that they can be converted to low-volatility aerosol precursors on sub-second timescales following a single oxidant attack. The α-pinene + $O_3$ system is particularly efficient in this regard. However, the actual mechanism behind this conversion is not understood. The key challenge is the steric strain created by the cyclobutyl ring in the oxidation products. This strain hinders subsequent unimolecular hydrogen-shift reactions essential for lowering volatility. Using quantum chemical calculations and targeted experiments, we show that the excess energy from the initial ozonolysis reaction can lead to novel oxidation intermediates without steric strain, allowing the rapid formation of products with up to 8 oxygen atoms. This is likely a key route for atmospheric organic aerosol formation.

[1] Aerosol Physics Laboratory, Tampere University, Tampere FI-33101, Finland. [2] Department of Chemistry, University of Helsinki, P.O. Box 55, Helsinki FI-00014, Finland. [3] Tomsk State University, 36 Lenin Avenue, Tomsk 634050, Russia. [4] Aerodyne Research, Inc., Billerica, MA 01821, USA. [5] Department of Atmospheric Science, University of Washington Seattle, Washington, WA 98195, USA. [6] Institute for Atmospheric and Earth System Research (INAR/Physics), University of Helsinki, P.O. Box 64, Helsinki FI-00014, Finland. [7] These authors contributed equally: Siddharth Iyer, Matti P. Rissanen. ✉email: siddharth.parameswaraniyer@tuni.fi; matti.rissanen@tuni.fi; theo.kurten@helsinki.fi

Aerosol affects the planet's radiation budget, especially by affecting the formation and properties of clouds, and consequently the climate. However, uncertainties remain high in estimates of the radiative effects of aerosol[1–3]. Epidemiological studies also show a clear link between high ambient aerosol concentration and increased mortality[4,5]. Organic carbonaceous matter typically comprises 20–90% of the aerosol mass in the lower troposphere[6], and improving our understanding of the impact of aerosol on both climate and health therefore requires a detailed characterization of the sources and fates of organic aerosol mass. Secondary organic aerosol mass (SOA) is formed in situ through the oxidation of gas-phase volatile organic compounds (VOC). The oxidation process transforms VOC into less volatile products, which partition into aerosol[7–9], or in the case of the lowest-volatility products, even participate in new-particle formation. Recently, some VOC have been shown to undergo autoxidation[10], a series of sequential peroxy radical ($RO_2$) hydrogen shifts (H-shifts) and $O_2$ additions to the hydrocarbon chain, leading to so-called highly oxygenated organic molecules (HOM) with very-low volatilities extremely rapidly[9,11,12]. While the fastest VOC to HOM conversion hitherto measured has a reaction time of around 1.5 s[12], if sufficiently rapid $RO_2$ H-shift channels are available, autoxidation could allow some VOC to HOM conversion to occur even on sub-second timescales following a single oxidant attack under natural temperature and pressure (NTP) conditions. Formation of low-volatile HOM that contribute to aerosol formation and growth from a single bimolecular reaction of a VOC precursor with $O_3$ is the organic analog to the formation of sulfuric acid, which has already for over a century been known as the most important inorganic aerosol precursor, largely due to its efficient gas-phase formation via a single rate-limiting bimolecular reaction of volatile $SO_2$ with the OH radical[13,14].

Monoterpenes ($C_{10}H_{16}$) account for ~11% of the global biogenic VOC emission by mass, and contribute significantly to the global SOA budget[15,16] because their oxidation products can have very low volatility. $\alpha$-pinene is the most emitted monoterpene, making up ~34% of the total monoterpene emission[17]. It is a very efficient SOA precursor[18], and the $\alpha$-pinene + $O_3$ reaction in particular is one of the main SOA-forming systems in the atmosphere[19–23]. This reaction has therefore been extensively studied over the past decades[18,24–30]. However, no reaction pathway leading unambiguously from $\alpha$-pinene ozonolysis to HOM has been reported to date.

The conventional understanding of the initial steps of the $\alpha$-pinene + $O_3$ reaction is illustrated in Supplementary Fig. 1. Straight chain alkenes, like ethene, have an additional diradical pathway, which is not relevant to cyclic alkenes, such as $\alpha$-pinene (see Supplementary Methods). Briefly, the conventional mechanism involves the formation of four possible structural isomers of so-called Criegee intermediates (carbonyl oxides; CI), which then undergo rapid unimolecular reactions to form either vinyl hydroperoxides (VHP; in three of four cases), or a dioxirane. Further reactions of the VHP lead, via vinoxy radical intermediates, to four different peroxy radical ($RO_2$) isomers. One of the $RO_2$ forming pathways is illustrated in detail in Fig. 1. This well-established reaction mechanism can unfortunately not explain the observed rapid SOA formation from $\alpha$-pinene + $O_3$, as H-shifts in all of the four $RO_2$ isomers are hindered by the steric strain created by the intact cyclobutyl (CB) ring[31]. Due to this strain, the barrier for the most favorable H-shift reaction is several kcal/mol higher than that of the corresponding H-shift in the cyclohexene + $O_3$ system (which lacks a CB ring)[11,31]. The slow H-shift rates limit autoxidation, and therefore the rate of VOC to HOM conversion. Kurtén et al. considered an isomerization reaction available to the vinoxy intermediate that leads

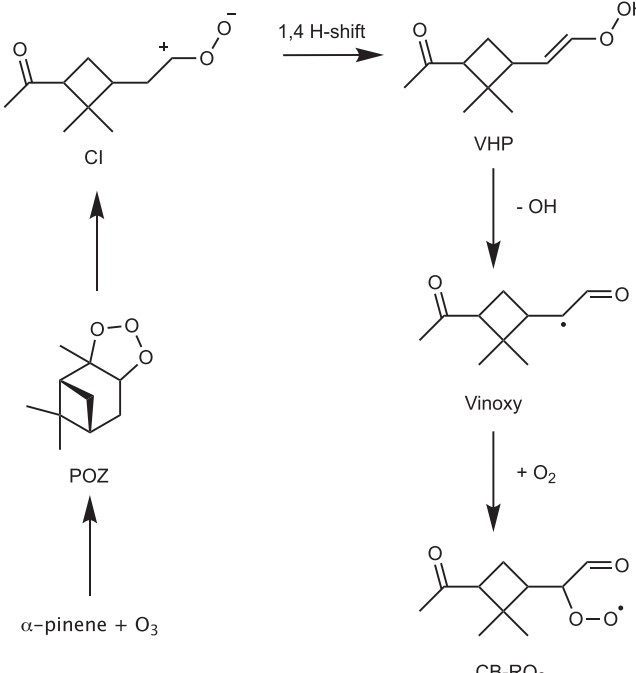

**Fig. 1 Ozonolysis reaction of $\alpha$-pinene.** The addition of ozone to the double bond creates a primary ozonide (POZ). This breaks apart into a Criegee intermediate (CI; one out of four shown here). The CI can undergo a 1,4 H-shift, forming a vinyl hydroperoxide (VHP). The VHP will rapidly lose an OH radical and form a vinoxy radical, which then adds an $O_2$ to form a peroxy radical with an intact cyclobutyl ring (CB-$RO_2$). Black dots indicate radical centers.

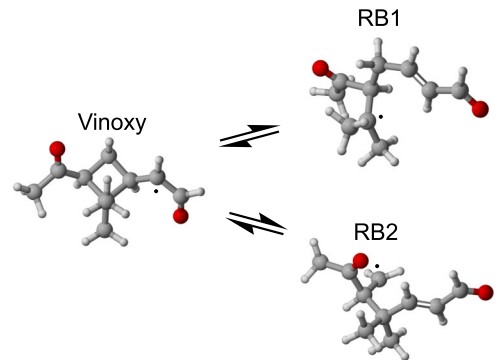

**Fig. 2 Ring-breaking isomerization reactions of the vinoxy radical shown in Fig. 1.** These reactions lead to alkyl radicals RB1 and RB2. Solid black circles indicate the atoms with radical centers. Color: carbon = gray, oxygen = red, hydrogen = white.

to a ring broken (RB) product (shown in Fig. 2), but found that it does not compete with $O_2$ addition under NTP conditions[31].

In this work, we show that excess energy from the initial ozonolysis of $\alpha$-pinene can activate otherwise inaccessible reaction pathways that lead to novel first-generation $RO_2$ without steric strain. Specifically, we model the effect of excess energy on the isomerization pathways available to one of the vinoxy intermediate isomers (see Fig. 2). Another ring-breaking isomerization reaction available to some CI isomers is also studied using multireference methods, as previously used computational methods have proved unreliable for describing this channel. Master equation calculations are carried out to estimate the fraction of RB products formed as a function of temperature and pressure.

Finally, flow-tube experiments of α-pinene ozonolysis coupled to a nitrate-based chemical ionization mass spectrometer (nitrate-CIMS) are conducted to confirm that the formation of HOM does indeed occur at sub-second timescales.

## Results

The quantum chemically calculated reaction stationary points (minima and saddle points along the potential energy surface = PES) of the studied pathway of α-pinene ozonolysis are shown in Fig. 3. The initial ozone addition to α-pinene has a barrier of 6.6 kcal/mol. The overall exothermicity of the vinoxy formation is 81.1 kcal/mol, which is partitioned into the vinoxy and the spectator component OH. All unimolecular transition states leading to its formation are below the reactants in energy. The isomerization of the vinoxy radical to RB1 and RB2 have barriers corresponding to transition states TS4 and TS5, depicted with red and yellow traces in Fig. 3, respectively, and the products of these reactions will add $O_2$ to form peroxy radicals RB1-$RO_2$ and RB2-$RO_2$, respectively. These two peroxy radicals were included in the master equation simulation because the reverse barriers going from the ring broken RB1 and RB2 to the ring intact vinoxy were lower than the barriers for their formation, and it was important to account for this reforming of the ring to accurately simulate the fraction of the RB $RO_2$ produced. The direct ring-breaking isomerization of the CI to CI_RB in Fig. 3 was found to be endothermic by 29 kcal/mol, and has a barrier (TS6 in Fig. 3) of 44.1 kcal/mol (both relative to the energy of the CI). Optimized geometries along the CI ring-breaking isomerization channel and details of the method used are provided in Supplementary Methods.

The fractional populations of the products CB-$RO_2$ and RB1-$RO_2$ as a function of the precursor α-pinene concentration when 50% of the reaction exothermicity is partitioned into the density of states of the vinoxy (and 50% into the spectator component OH; details are provided in the following discussion section and in Supplementary Note 1) are shown in Fig. 4. The fractional populations of the products when the reaction exothermicity is unequally partitioned between the vinoxy and the OH, as well as other sensitivity tests indicating that the qualitative results are robust with respect to the likely error margins of the computational methods, are provided in Supplementary Notes 1 and 2.

## Discussion

Ideally, trajectory calculations are needed to accurately predict the amount of the available excess energy distributed to the rovibrational modes of the vinoxy and the OH following the decomposition of the VHP intermediate. Since these are computationally too expensive to carry out at a sufficiently high level of theory, we instead controlled the fraction of the excess energy distributed to the vinoxy by introducing a parameter $m$ (between 0 and 1) to be applied to the density of states of the vinoxy during the master equation simulation (see Supplementary Note 1). This procedure has been followed previously by Shannon et al. for a similar study on the fate of activated bimolecular products—the significantly exothermic glyoxal + OH reaction[32]. Due to the high barrier involved, the isomerization of the CI to ring broken CI_RB does not occur. However, a significant fraction of RB products produced via the vinoxy channel was observed. The vinoxy preferentially isomerizes to RB1 instead of RB2 because of the relatively lower barrier for the former. The product fraction of RB1-$RO_2$ reached a maximum of ~82% for $m = 0.45$ and remained unchanged for higher $m$ values. However, for $0.05 \leq m \leq 0.4$, the RB1-$RO_2$ fraction was reduced to between 33 and 81%, with a reciprocal increase in CB-$RO_2$. For $m = 0$, only 12% of RB1-$RO_2$ was observed, which is expected as the density of

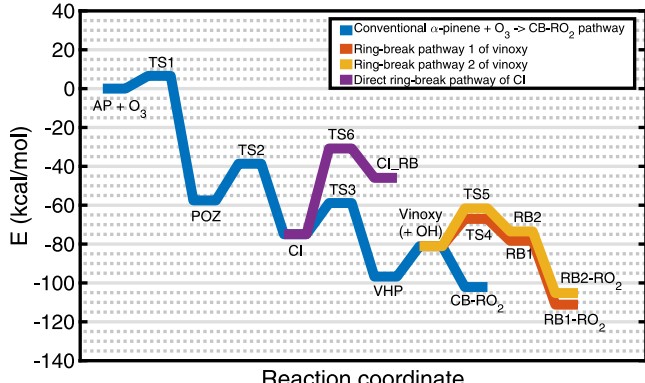

**Fig. 3 Stationary points along the PES of α-pinene ozonolysis reaction.** Zero-point corrected energies are shown on the y-axis and the reaction coordinate on the x-axis. AP = α-pinene, TS = transition state, CI = Criegee intermediate, POZ = primary ozonide, VHP = vinyl hydroperoxide, CB = cyclobutyl, RB = ring-broken intermediate, $RO_2$ = peroxy radical. Source data are provided as a Source Data file.

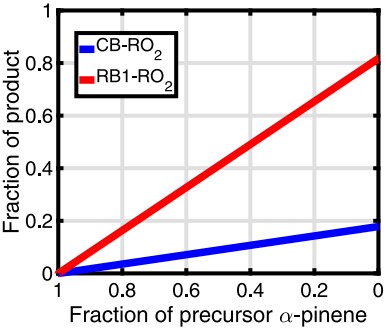

**Fig. 4 Fraction of products cyclobutyl CB-$RO_2$ and ring-broken RB1-$RO_2$.** The product fractions are shown as a function of α-pinene consumed when 50% of the reaction exothermicity is partitioned into the vinoxy co-product and 50% into the spectator OH co-product. $T = 298$ K and $P = 760$ Torr. Source data are provided as a Source Data file.

states of the vinoxy is replaced by a single classical harmonic oscillator in this extreme case. Including the hindrance potentials for the internal rotations for the intermediates and TSs along the PES (excluding the TS6 and CI_RB as this channel was found to be unimportant) in the simulation increased the RB1-$RO_2$ fraction to ~89% (see Supplementary Note 2), indicating that the effect is small, likely due to the very large number of vibrational modes of the intermediates and products along the PES. Assuming that the four Criegee-forming channels of α-pinene + $O_3$ have roughly equal yields, the 89% yield of RB1-$RO_2$ from the single channel studied here translates into a total yield of RB products of about 22% of RB1-$RO_2$ for the overall α-pinene ozonolysis reaction. It should be noted that the ~11% production of CB-$RO_2$, which is formed from the thermalized fraction of the intermediates along the PES, is in qualitative, and very nearly quantitative agreement with the experimentally observed yield of 15% (±7%) of stabilized CI in α-pinene ozonolysis[33].

The unimolecular H-shift pathways available to CB-$RO_2$ are very slow[31]. In contrast, RB1-$RO_2$ contains a double bond, and can consequently form an endoperoxide. Based on a comparison to similar reactions in the related α-pinene + OH system, this reaction likely has a high rate coefficient (around or above $1 \, s^{-1}$)[34]. RB1-$RO_2$ could also undergo an aldehydic H-abstraction, but this was found to be too slow to be competitive with the endoperoxide reaction (details in Supplementary Note 5). The endoperoxide reaction leads to a radical center on the adjacent carbon, which

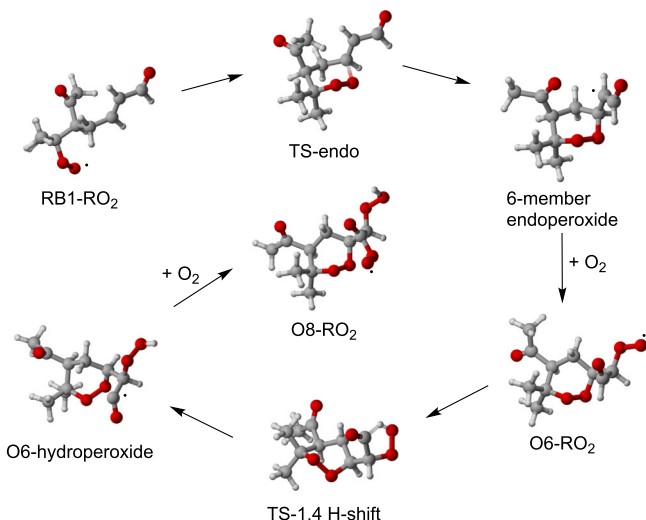

**Fig. 5 Autoxidation steps available to ring-broken RB1-RO₂.** The peroxy radical can form a six-member endoperoxide via the transition state TS-endo, which subsequently adds an O₂ to form a six-oxygen containing peroxy radical O6-RO₂. This molecule can undergo a fast 1,4 aldehydic H-shift to form O6-hydroperoxide via the transition state TS-1,4 H-shift, which subsequently adds an O₂ to form an eight-oxygen containing peroxy radical O8-RO₂.

subsequently adds an O₂ molecule. The reverse reaction of endoperoxide ring-opening (reforming a peroxy radical) is known to not be competitive with O₂ addition[34]. Depending on the carbon the O₂ adds to, the endoperoxide forms as either a 6-member or a seven-member ring. The former is likely around two orders of magnitude faster[34]. The isomerization of RB1-RO₂ to a six-member ring endoperoxide was thus found to be the competitive channel (see Supplementary Note 5). Following the endoperoxide reaction and subsequent O₂ addition, the peroxy radical will likely undergo a 1,4 H-shift, abstracting the aldehydic hydrogen atom. This is a fast reaction, with rates in excess of $1\,s^{-1}$[35]. The O₂ addition following the aldehydic H-shift leads to the peroxy radical $C_{10}H_{15}O_8$. Thus, starting from RB1-RO₂, one endoperoxide formation step and one aldehydic H-shift step, both rapid, can quickly form an eight-oxygen containing HOM RO₂. The steps are shown in Fig. 5.

The presence of an endoperoxide group in $C_{10}H_{15}O_8$ is supported by earlier experiments of α-pinene ozonolysis with an added D₂O flow. The H-to-D exchange in the observed mass spectra indicates the number of exchangeable H-atoms (in practice, –OH or –OOH groups) in the molecule. $C_{10}H_{15}O_8$ was found to undergo only a single H-to-D exchange, indicating only one H-shift reaction, thus necessitating other propagation pathways, likely endoperoxide formation[27]. This is a strong experimental indicator that a fraction of the initial RO₂ following α-pinene ozonolysis contains a double bond—most likely the completely ring-broken RB1-RO₂. The rapidity of the formation of the $C_{10}H_{15}O_8$ peroxy radical was explored in a laboratory setting for precursor residence times between 75 and 1000 ms using a nitrate-CIMS. To the best of our knowledge, this is the first time that such rapid autoxidation reaction times were studied using a CIMS instrument, and this was enabled by the design of the multi-scheme chemical ionization inlet (MION) employed in this work[36]. To obtain the initial RO₂ within the quick residence time of our experiments, high loading conditions were employed. While some bimolecular RO₂ reactions are possible under these short reaction time conditions, multiple bimolecular RO₂ reactions are likely negligible. This was confirmed by running kinetic modeling simulations with realistic precursor concentrations and literature rate coefficients (see Supplementary Note 7). Mass

spectra for α-pinene ozonolysis at residence times 75 and 300 ms are shown in Fig. 6. The $C_{10}H_{15}O_8$ peroxy radical was observed to form extremely rapidly—within 75-ms residence time, and the next ten-oxygen containing RO₂ inside 300 ms. Such rapid formation of these higher oxidized RO₂ necessitates that they no longer have an intact CB ring, and likely follow the endoperoxide pathway described previously.

We show in this work that the excess energy from α-pinene ozonolysis opens up novel reaction channels. About 89% of first-generation RO₂ from one of the four possible ozonolysis pathways are completely RB. This explains the hitherto mysterious sub-second formation of highly oxidized organic products in one of the most important atmospheric SOA-forming systems.

## Methods

**Electronic structure calculations.** Systematic conformer sampling of reactants, intermediates, transition states, and products studied here were performed using Spartan'14[37] and Spartan'18[38] programs. The MMFF method was used to carry out the conformer sampling. For non-transition state molecules, all conformers were first optimized at the ωB97X-D/aug-cc-pVTZ level of theory using the Gaussian 09 program[39]. For the transition states, the approximate TS structure was first built in Spartan, and then optimized at the B3LYP/6-31+G(d) level in Gaussian 09 with the relevant TS bond distances constrained. The B3LYP functional was used for the initial TS calculations as it has been shown to work well in finding TS geometries[34]. Following the constrained optimization of the approximate TS geometry, unconstrained TS optimization was carried out at the same level of theory. Once the TS was found, the geometry was taken back to Spartan for conformer sampling with the relevant bonds constrained. The multiple TS conformers were once again optimized with constraints, and subsequently run through a unconstrained TS optimization at the B3LYP/6-31+G(d) level in Gaussian 09. Conformers within 2 kcal/mol in relative electronic energies were optimized at the higher ωB97X-D/aug-cc-pVTZ level. Single point electronic energy calculation was carried out at the ROHF-ROCCSD(T)-F12a/VDZ-F12 level for the lowest energy conformers of both the transition state and non-transition state structures using the Molpro program[40].

For the Criegee pathway, preliminary calculations using the SS-CASSCF(8,6)/6-311++G(d,p) method showed that the wavefunction of the transition state of the Criegee ring-break channel has a multireference character, with two dominant determinants (with weights of −0.78 and 0.56 in the configuration interaction expansion). A similar situation exists for the product of this channel, where the ground state is best described as an open shell singlet. In both cases, multireference methods are required to treat them accurately. Preliminary calculations on the vinoxy ring-break channel using the SS-CASSCF(7,6)/6-311++G(d,p) method showed that the wavefunctions of the transition state, the reactant, and the product of this channel are all single-reference, with the main determinant having a weight of at least 0.94. However, in order to compare the activation barriers of the Criegee and vinoxy pathways correctly, the same method with similar options should be used. Therefore, we applied XCM-QDPT2 with the same active space and basis set for both pathways (accounting for the loss of one "active" electron to the OH radical in the VHP dissociation reaction forming the vinoxy). We chose the eight electrons, six orbital active space for the configuration interaction, which corresponds to a seven electron, six orbital active space in the vinoxy radical. We chose this active space because it describes both reactions fairly well, and is quite stable, without any rotations or swapping of molecular orbitals in the optimization procedure. The active orbitals for the CI and vinoxy radical are shown in Supplementary Tables 1 and 2, respectively.

**Master equation calculations.** The Master equation solver for multi-energy well reactions (MESMER) program[41] was used to carry out the RRKM simulations. The initial association reaction was treated using the SimpleRRKM method in MESMER with Eckart tunneling. The other intermediate complexes separated by transition states were also similarly treated. The nascent energy distributed to the vinoxy and OH products from VHP decomposition were approximated using the pseudo-isomerization methodology[32]. The O₂ addition reactions to vinoxy/alkyl radicals were treated using the "Simple Bimolecular Sink" method in MESMER with a bimolecular loss rate coefficient of $2 \times 10^{-12}$ cm³ molecule⁻¹ s⁻¹ and with an O₂ "excess reactant" concentration of $5 \times 10^{18}$ molecules cm⁻³. All intermediates were assigned as "modeled" in the simulations and given Lennard–Jones potentials sigma = 6.5 Å and epsilon = 600 K. These are identical to those used by Kurtén et al. for their α-pinene and Δ₃-carene systems[42]. MESMER utilizes the exponential down ($\Delta E_{down}$) model for simulating the collisional energy transfer. For N₂ bath gas, the MESMER recommended values for $\Delta E_{down}$ are between 175 and 275 cm⁻¹, and a value of 225 cm⁻¹ was used in our simulations. In addition, a grain size of 100 and a value of 60 $k_B T$ for the energy spanned by the grains were used. Ozone was set as the excess reactant and given a high value of $1 \times 10^{18}$ molecules cm⁻³. While not relevant to ambient conditions, the high ozone value allowed for rapid formation of products and changes in species profiles in our

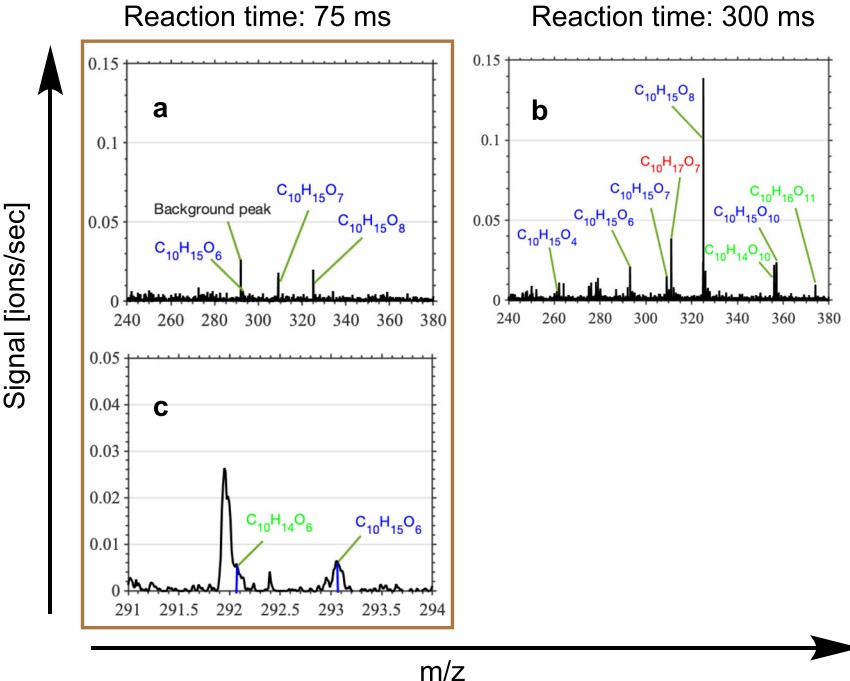

**Fig. 6 Nitrate-CIMS spectra for α-pinene ozonolysis.** The experiments carried out at reaction times 75 ms (**a, c**) and 300 ms (**b**) indicate rapid formation of peroxy radical $C_{10}H_{15}O_8$. The background peak at 75 ms in **a** is expanded in **c**. Blue labels indicate radical and green labels indicate closed-shell species derived from α-pinene ozonolysis. Red label indicates possible radical species from OH oxidation of α-pinene.

simulations. Test calculations were run with the more ambiently relevant $[O_3] = 1 \times 10^{12}$ molecules $cm^{-3}$, and the ratio between $CB\text{-}RO_2$ and $RB1\text{-}RO_2$ fraction populations at the end of the simulation time was near identical to those at the higher $[O_3]$ values. This makes sense since, apart from the initial association reaction, the rest of the ozonolysis steps until the formation of the vinoxy radical are unimolecular. While the formation of the POZ does indeed depend on the initial ozone concentration, the remaining steps do not, and their rates remain the same for the different $[O_3]$ values. Giving a high initial ozone concentration thus allowed us to simulate the interplay between product fractions as a function of time, while keeping the simulation time (and therefore the computation time) feasible. See Supplementary Methods for more details.

**Chemical ionization time-of-flight mass spectrometry.** A nitrate-based high-resolution time-of-flight chemical ionization mass spectrometer (nitrate-CIMS) was used to detect the products of α-pinene ozonolysis. α-pinene was bubbled from a liquid reservoir using an $N_2$ flow, and ozone was generated by flowing synthetic air through an ozone generator fitted with a 184.9-nm (Hg PenRay) lamp. The two precursors were allowed to react in the presence of $O_2$ in a quartz tube reactor and the products were subsequently detected. The experiments were conducted under atmospheric conditions. The inlet flow rate was set to either 20 or 30 l per minute to achieve rapid sub-second reaction times, which was made possible by the MION[36] used in this work.

## Data availability

Source data are provided with this paper. The ab initio output files (.log and .out) and the MESMER files that support the findings of the manuscript are provided at: https://doi.org/10.5281/zenodo.4297276. The mass spectrometry data are available upon request from the corresponding authors. Source data are provided with this paper.

## Code availability

The scripts used to produce the plots are provided with this paper.

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

## Acknowledgements
This study has received funding from the Academy of Finland (331207, 1315600). We acknowledge Dr Struan Robertson (Dassault Systèmes) for MESMER support. We thank the TofTools team for providing tools for mass spectrometry analysis, and the CSC IT Center for Science in Espoo, Finland, for providing the computing resources.

## Author contributions
M.P.R., T.K., M.E., and J.T. devised the research. S.I and R.V. carried out the electronic structure and master equation calculations; M.P.R. and J.E.K. designed the experimental setup, S.I., S.B., and M.P.R. carried out the experiments, and S.I. analyzed the data. S.I. wrote the paper with contributions from all co-authors.

## Competing interests
The authors declare no competing interests.
