## [Peer Review File · Nature Communications]

REVIEWER COMMENTS

Reviewer #1 (Remarks to the Author):

The authors explore the feasibility of a novel pathway for conversion of alpha-pinene to highly oxidized organic molecules (HOM) through a combination of state-of-the-art theoretical, experimental, and modeling analyses. The final result is a very valuable indication of an important new route to forming HOM in the atmospheric. The methodologies and assumptions are well stated and the conclusions are well validated. The sensitivity analysis in particular is quite informative. Thus, I recommend that this manuscript be accepted for publication in Nature Communications.

I have a few minor suggestions for improving the manuscript:

With multi reference calculations there are often multiple ways in which the active orbitals are selected. Thus, it would be helpful if the authors provided an explicit description of how their (8,6) active space correlates to specific orbitals in the molecule.

The comparison in Table S3 should be updated to include the most accurate prior estimates of the $C_2H_4 + O_3$ energies. For example, the study of Pfeifle et al. [J. Chem. Phys. 148, 174306 (2018)] provides higher accuracy predictions than any of those included in the Table. The authors should also check to be sure that there are not other even more recent and higher accuracy studies.

In ethylene ozonolysis there is also a diradical pathway that leads to a ketohydroperoxide (cf. Fig. 1 of Pfeifle et al.). This diradical pathway carries significant flux. I am curious what is known about the significance of any related pathway for the ozonolysis of alpha-pinene. The authors make no mention of it in their preliminary review of pathways. If it was present, it could provide an alternative to that studied in the present work, as it is 50 kcal/mol more exothermic than the CI pathway. With this excess energy, it might also be quite feasible to break the ring.

The Pfeifle et al. study also explores direct trajectory simulations of the energy redistribution in ozonolysis. The authors may wish to be aware of that.

On the first page of section S2, the authors describe a rate coefficient as having a magnitude of $2 \times 10^{12} \text{ cm}^3 \text{ molecule}^{-1} \text{ s}^{-1}$. There must be a typo in the exponent of this expression. Perhaps it should be 10^{-12} .

The citations at the end of the Supplementary material are messed up. I., e., they are in the order, 1, 2, 3, 4, 3, 5, ...

Reviewer #2 (Remarks to the Author):

This is a nice piece of work touching on two key areas of emerging importance in atmospheric gas phase chemistry, namely the formation mechanism of HOM and the importance of non-thermal, partially stabilised intermediates. The article is well written and clearly sets out the important background and the implications of the current work. The supplementary information is very detailed and I welcome the printing of input files such that other can replicate these results. In that vein it would be nice to see some more details on their multireference calculations, specifically regarding the orbitals they choose to use in the active space.

The authors rightly point out that energy partitioning is generally dynamical rather than statistical in nature so take the pragmatic approach of varying the partition of energy in the fragments by hand. This coupled with the experimental support gives strong evidence that the proposed channel is operational although the results are qualitative rather than quantitative. The limiting value of 0.82 RB1_RO2 is presented in a more quantitative light, however it appears that the authors have not treated any hindered rotations in the vinoxy or the TS and this leads to a great deal of uncertainty regarding the kinetics. I would like to see the authors include the hinderance potentials in the vinoxy and the relevant transitions states to firm up the limiting value.

In summary I think this a nice paper and worthy of inclusion in nature communications. The theory is qualitative and pragmatic but mostly open about this and is well supported by the experimental results to make a strong combined case for the importance of the proposed chemistry. If they address the issue of hindered rotors above and add a little more information on multireference methods, then I would strongly support publication.

Reviewer #3 (Remarks to the Author):

Review of “Molecular mechanism for rapid autoxidation in α -pinene ozonolysis” by Iyer et al.
(NCOMMS-20-39478-T)

This paper describes the theoretically- and experimentally-supported proposal of new pathways leading the rapid formation of highly oxygenated organic molecules (HOM) with 8 oxygen atoms in α -pinene ozonolysis. HOM from the oxidation of monoterpenes are known to be key species of the new particle formation in boreal forests, which influences on the number of cloud condensation nuclei (CCNs) and then affects earth’s climate. It was reported that HOM in α -pinene ozonolysis were produced rapidly (sub-second timescales), but the formation mechanism of the HOM is not fully understood. Indeed, it was believed that they were produced via the repetition of autoxidation, but H-shifts were found to be hindered by the steric strain caused by the cyclobutyl ring which α -pinene has originally. I think that this is a mystery in nature and that readers in broader research fields would be interested in the topics. In this work, it was theoretically shown that excess energy from the initial α -pinene ozonolysis can lead to ring breaking isomerization reactions of vinyloxy radicals formed in vinyl hydroperoxide channels and that HOM with 8 oxygen atoms can be produced rapidly from the first-generation peroxy radicals (ring-breaking vinyloxy radical-O₂) without the steric strain. In addition, the formation of the HOM with 8 oxygen atoms within several tens of milliseconds was confirmed by NO₃⁻ chemical ionization mass spectrometry (NO₃⁻-CIMS). This paper is generally well-organized and well-written, but I think that there are two weak points in this paper.

The first point is that the proposed ring-breaking of vinyloxy radicals follows the dissociation of OH and vinyloxy radical with the cyclobutyl ring. I wonder if a large amount of the excess energy from the initial α -pinene ozonolysis would be lost as translational energy at the dissociation. In addition, I wonder if the vinyloxy radical having the cyclobutyl ring with some internal energy would undergo quenching by air with possibly microsecond timescales. Indeed, it is reported that the yields of stabilized Criegee intermediates and OH radicals are 0.15 ± 0.07 and $0.8-0.9$, respectively, in α -pinene ozonolysis. Thus, the quenching by air partly occurs. So, can the authors estimate an absolute yield of the ring-breaking vinyloxy radicals from the present theoretical calculations?

The second is that the authors did not show temporal variations of C₁₀H₁₅O₄ and C₁₀H₁₅O₆ in Figure 6. I guess that those radicals were also detected by NO₃⁻-CIMS, by which the detection sensitivities of C₁₀H₁₅O₄, C₁₀H₁₅O₆, and C₁₀H₁₅O₈ were similar, and that the ion signals of C₁₀H₁₅O₄ and C₁₀H₁₅O₆ should be observed according to Figure S14.

I think that this paper is worth for publication in Nature Communications. I would like to recommend the authors to reinforce the present conclusions, considering my above-mentioned comments, before it is acceptable for publication. And some minor comments are listed as follows.

Minor comments:

(1) Page 2, Lines 27–28: With regard to the name of HOM, “highly oxygenated organic molecules” is called as HOM in a review paper by Bianchi et al. (2019), in which many authors in the present paper are co-authors. In order for readers not to be confused, it is better to use “highly oxygenated organic molecules” here.

(2) Page 4, Line 76: section S6 → section S7

(3) Page 6, Line 14 of the Supplementary information: 2×10^{12} → 2×10^{-12}

(4) Page 16, Line 10 of the Supplementary information: equation ES.2 → equation S7.2

(5) Page 16, Line 14 of the Supplementary information: equation S7.2 → equation S7.1 (Am I correct?)

(6) S10 of the Supplementary information: There are two ref. 3. Kurten et al. (2017) should be deleted. The information of ref. 11 should be updated.

References:

Bianchi, F. et al. Highly Oxygenated Organic Molecules (HOM) from Gas-Phase Autoxidation Involving Peroxy Radicals: A Key Contributor to Atmospheric Aerosol. *Chem. Rev.* 2019, 119, 3472–3509.

We thank all three reviewers for their valuable input that helped improve the manuscript.

Reviewer 1:

1. With multi reference calculations there are often multiple ways in which the active orbitals are selected. Thus, it would be helpful if the authors provided an explicit description of how their (8,6) active space correlates to specific orbitals in the molecule.

Author comment: We absolutely agree. Additional multireference description is now provided in section S1.2 of the supplementary file.

Changes: Additional description of the multireference calculations has been added to section S1.2.

2. The comparison in Table S3 should be updated to include the most accurate prior estimates of the $C_2H_4 + O_3$ energies. For example, the study of Pfeifle et al. [J. Chem. Phys. 148, 174306 (2018)] provides higher accuracy predictions than any of those included in the Table. The authors should also check to be sure that there are not other even more recent and higher accuracy studies.

Author comment: Thank you for the reference. The energetics of the $C_2H_4 + O_3$ reaction from the Pfeifle et al. study has been added to Table S3. We were unable to find a more recent higher accuracy study.

Changes: Table S3 now includes the $C_2H_4 + O_3$ energies from the Pfeifle et al. study.

3. In ethylene ozonolysis there is also a diradical pathway that leads to a ketohydroperoxide (cf. Fig. 1 of Pfeifle et al.). This diradical pathway carries significant flux. I am curious what is known about the significance of any related pathway for the ozonolysis of alpha-pinene. The authors make no mention of it in their preliminary review of pathways. If it was present, it could provide an alternative to that studied in the present work, as it is 50 kcal/mol more exothermic than the CI pathway. With this excess energy, it might also be quite feasible to break the ring.

Author comment: The transition state in the step-wise diradical pathway described in Pfeifle et al. requires an internal C-C rotation in either direction to avoid minimizing to the Criegee forming TS. Such a C-C rotation is severely hindered in the α -pinene POZ system due to the 6-member ring, and we therefore believe that the diradical channel is not relevant to our study. A description of this channel has nevertheless been added to section S1.1 in the supplementary file.

Changes: A description of the diradical pathway from alkene ozonolysis has now been added to page 3 line 49 in the manuscript and in section S1.1 in the supplementary file.

4. The Pfeifle et al. study also explores direct trajectory simulations of the energy redistribution in ozonolysis. The authors may wish to be aware of that.

Author comment: Thank you. This was an extremely helpful reference.

Changes: -

5. On the first page of section S2, the authors describe a rate coefficient as having a magnitude of $2 \times 10^{12} \text{ cm}^3 \text{ molecule}^{-1} \text{ s}^{-1}$. There must be a typo in the exponent of this expression. Perhaps it should be 10^{-12} .

Author comment: Thank you for pointing this out. It is indeed 10^{-12} and has been corrected in section S2.

Changes: Section S2 in the supplementary file: "...MESMER with a bimolecular loss rate coefficient of $2 \times 10^{12} \text{ cm}^3 \text{ molecule}^{-1} \text{ s}^{-1}$..." => "...MESMER with a bimolecular loss rate coefficient of $2 \times 10^{-12} \text{ cm}^3 \text{ molecule}^{-1} \text{ s}^{-1}$..."

6. The citations at the end of the Supplementary material are messed up. I., e., they are in the order, 1, 2, 3, 4, 3, 5, ...

Author comment: The citation order has now been corrected.

Changes: The citation order in the supplementary file has been corrected.

Reviewer 2:

1. This is a nice piece of work touching on two key areas of emerging importance in atmospheric gas phase chemistry, namely the formation mechanism of HOM and the importance of non-thermal, partially stabilised intermediates. The article is well written and clearly sets out the important background and the implications of the current work. The supplementary information is very detailed and I welcome the printing of input files such that other can replicate these results. In that vein it would be nice to see some more details on their multireference calculations, specifically regarding the orbitals they choose to use in the active space.

Author comment: Thank you! We note also that the optimized coordinates of the molecules studied can be extracted from the example MESMER input file in section S9 in the Supplementary, should another group wish to repeat our quantum chemical calculations, or redo them with another method. In addition, the output files are also freely available in a data archive. The product fraction values as a function of precursor concentrations during an example MESMER run are provided in a supplementary excel file. The requested details regarding the orbital choice in the multireference calculations have been added to section S1.2 of the supplementary file.

Changes: Details of our multireference calculations have been added to section S1.2 of the supplementary file.

2. I would like to see the authors include the hinderance potentials in the vinoxy and the relevant transitions states to firm up the limiting value.

Author comment: Thank you for pointing this out. We have now carried out an additional MESMER simulation with the hinderance potentials included for all the intermediates and TSs along the PES (excluding the CI ring-break TS and product) to estimate the final product fractions. The results have been added to the "sensitivity" section S3 in the supplementary file. These simulations indicate that the effect of including internal rotations is small – probably due to the very large number of vibrational modes (81 for systems from the POZ to the VHP; 75 for the vinoxy to

RB1/RB2, and again 81 for CB-RO₂, RB1-RO₂ and RB2-RO₂), as well as the fact that the number of internal rotational modes changes fairly little across the potential energy surface from the CI onwards (i.e. much of their effects are cancelled out as many of the same rotors are present in the reactants, transition states and products).

Changes: A description of the methods used to treat hindered rotors and the results of the subsequent MESMER simulation have been added to section S3 in the supplementary file and mentioned in page 9 line 125 in the manuscript.

Reviewer 3:

1. The first point is that the proposed ring-breaking of vinyoxy radicals follows the dissociation of OH and vinyoxy radical with the cyclobutyl ring. I wonder if a large amount of the excess energy from the initial α -pinene ozonolysis would be lost as translational energy at the dissociation.

Author comment: We agree that the excess energy lost to translational energy is an important issue, which is fortunately simulated (at least to some extent) by the MESMER program. The pseudo-isomerization methodology in MESMER that was used to treat subsequent unimolecular reactions of excited product fragments (vinyoxy and OH in our case) partitions the available excess energy into the rovibrational modes of the products and into their relative translational energy (see equations S1 and S2 in the supplementary file). “Loss” of excess energy into translational modes is thus considered when MESMER calculates the final product fractions. Pfeifle et al. 2018 carried out trajectory calculations on the ethene + O₃ system and found that for the C₂H₄ + O₃ => CH₂O₂ (Criegee intermediate) + HCOH reaction, 90% of the energy is partitioned into the relative translation energy of the two product fragments (Pfeifle et al. 2018). The vinyoxy intermediate in our system has tens of internal modes more than CH₂O₂ (75 vs 12 to be exact) and therefore more energy is expected to be partitioned into the rotational and vibrational modes of the vinyoxy. (If the relative percentage of energy going into each vibrational mode were roughly similar for the two systems, we would expect slightly less than half of the energy to partition into translational modes for our system). In our sensitivity tests, assigning zero internal modes of the vinyoxy to spread the excess energy (in effect replacing the density of states of the vinyoxy with that of a single classical harmonic oscillator) over still resulted in about 12% ring-broken peroxy radical product (Figure S6; m=0), indicating that some of the translational energy of the vinyoxy is converted to useful rotational and vibrational energy following collisions with the bath gas to lead to the ring-breaking isomerization.

Changes: A discussion of the trajectory calculation results from Pfeifle et al. 2018 in the context of our α -pinene ozonolysis system has been added to section S2 in the supplementary file.

2. In addition, I wonder if the vinyoxy radical having the cyclobutyl ring with some internal energy would undergo quenching by air with possibly microsecond timescales.

Author comment: The quenching of the vinyoxy radical, as well as the other intermediates along the PES, is explicitly considered by MESMER via collisions with a bath gas at a user specified pressure. In our simulations, we set the pressure at 1 atm (760 Torr) and used N₂ as the bath gas. The fraction of CB-RO₂ produced is an indication of the fraction of vinyoxy radicals that have lost enough of their internal energy to not undergo the ring-breaking isomerization reaction, but add an O₂ instead. This pressure dependence can be observed from Figure S8 where the fraction of CB-RO₂ goes up (and that of RB1-RO₂ correspondingly down) as the simulation pressure is increased.

3. Indeed, it is reported that the yields of stabilized Criegee intermediates and OH radicals are 0.15 ± 0.07 and 0.8-0.9, respectively, in α -pinene ozonolysis. Thus, the quenching by air partly occurs. So, can the authors estimate an absolute yield of the ring-breaking vinoxy radicals from the present theoretical calculations?

Author comment: The MESMER simulation considers the collisional quenching of all intermediates along the studied PES, and the final ring-broken RB1-RO₂ product fractions are indicative of the yield of ring-broken vinoxy. The lower reverse barrier for the reforming of the ring of the vinoxy (RB1 => vinoxy in Figure 3 in the manuscript) means that some of the ring-broken vinoxy is lost to this reverse reaction (our sensitivity test in section S3 in the supplementary file indicate this), which is why we extended the studied PES to the formation of the ring-broken peroxy radical RB1-RO₂ to derive an accurate estimate of ring-broken products from the particular ozonolysis channel studied here (one out of four in total).

We note that while quenching of the Criegee Intermediates is thus already modelled, our simulations do not contain other competitive (uni- or bimolecular) sinks for the CIs (as TS6 has been found to be unimportant), so also the thermalized CIs will form VHPs in our simulation. (It must be remembered that also the thermal reaction CI-to-VHP conversion is quite fast.) Quenching of the CI (along with the VHP and vinoxy and in principle – though not in practice – the POZ) contributes to lowering the fraction of ring-broken peroxy radicals formed. We also note that our predicted yield of ring-intact peroxy radicals (CB-RO₂; 11%) is not only qualitatively, but almost quantitatively, in agreement with the experimental result that about 15% of the Criegee Intermediates are thermalized (and thus contribute to the formation of CB-RO₂). This has now been noted in the discussion.

We are unable to confidently estimate a quantitative yield for RB1 from the overall ozonolysis process, as the yields four different Criegee Intermediates from the initial ozone addition reaction are unknown, and estimating them would likely require prohibitively expensive direct dynamics simulations. However, as the formation energetics of the different CIs are to our knowledge reasonably similar, we can crudely assume that each CI will be former at roughly equal yields, i.e. 25%. The overall yield of RB1 from the α -pinene + O₃ reaction would then be approximately 89% \times 25%, or about 22% according to our best guess, with a “worst-case” lower limit (corresponding to the m=0 simulation) of 12% \times 25% = 3%. Assuming the predicted autoxidation steps from RB1 onward are rapid, even this modest yield would be important for atmospheric SOA formation.

Changes: Description of the quenching of the Criegee Intermediate and an estimation of the yield of RB1-RO₂ has been added to section S3 in the supplementary file and in page 9 of the manuscript.

4. The second is that the authors did not showed temporal variations of C₁₀H₁₅O₄ and C₁₀H₁₅O₆ in Figure 6. I guess that those radicals were also detected by NO₃⁻-CIMS, by which the detection sensitivities of C₁₀H₁₅O₄, C₁₀H₁₅O₆, and C₁₀H₁₅O₈ were similar, and that the ion signals of C₁₀H₁₅O₄ and C₁₀H₁₅O₆ should be observed according to Figure S14.

Author comment: Thank you for pointing this out. The experimental spectra figure (Figure 6) in the main manuscript has been replaced with one that includes the C₁₀H₁₅O₄ and C₁₀H₁₅O₆ peaks. We note, however, that the detection sensitivities of C₁₀H₁₅O₄, C₁₀H₁₅O₆ and C₁₀H₁₅O₈ are likely not similar. Previous theoretical works have shown that the nitrate-CIMS method is not very sensitive to molecules with less than 6-oxygen atoms (Hyttinen et al. 2017, Hyttinen et al. 2018). This is because of the strong HNO₃*NO₃⁻ binding energy, and any molecule that will be efficiently detected by nitrate-CIMS will need to have a binding energy to NO₃⁻ that is higher than that of HNO₃ to NO₃⁻. The binding energy of α -pinene ozonolysis products to NO₃⁻ tends to increase with

an increase in the number of O atoms as it often corresponds to an increase in the number of hydrogen-bond donating groups.

Changes: New Figure 6 has been added to the manuscript which shows the extended nitrate-CIMS spectra at 75 ms and 300 ms residence time.

5. Minor comments:

- 1) Page 2, Lines 27–28: With regard to the name of HOM, “highly oxygenated organic molecules” is called as HOM in a review paper by Bianchi et al. (2019), in which many authors in the present paper are co-authors. In order for readers not to be confused, it is better to use “highly oxygenated organic molecules” here.
- (2) Page 4, Line 76: section S6 → section S7
- (3) Page 6, Line 14 of the Supplementary information: $2 \times 10^{12} \rightarrow 2 \times 10^{-12}$
- (4) Page 16, Line 10 of the Supplementary information: equation ES.2 → equation S7.2
- (5) Page 16, Line 14 of the Supplementary information: equation S7.2 → equation S7.1 (Am I correct?)
- (6) S10 of the Supplementary information: There are two ref. 3. Kurten et al. (2017) should be deleted. The information of ref. 11 should be updated.

Author comment: Thank you for pointing these out. All of these corrections have now been made.

Changes: The suggested corrections have been made to the manuscript and the supplementary file.

References:

Hyttinen, N.; Rissanen, M. P.; Kurtén, T. Computational comparison of acetate and nitrate chemical ionization of highly oxidized cyclohexene ozonolysis intermediates and products. *J. Phys. Chem. A* **2017**, *121*, 2172-2179.

Hyttinen, N.; Otkjaer, R.; Iyer, S.; Kjaergaard, H. G.; Rissanen, M. P.; Wennberg, P. O.; Kurtén, T. Computational comparison of different reagent ions in the chemical ionization of oxidized multifunctional compounds. *J. Phys. Chem. A* **2018**, *122*, 269-279.

Pfeifle, M.; Ma, Y.T.; Jasper, A.W.; Harding, L.B.; Hase, W.L.; Klippenstein, S.J. Nascent energy distribution of the Criegee intermediate CH₂OO from direct dynamics calculations of primary ozonide dissociation. *J. Chem. Phys.* **2018**, *148*(17), p.174306.

REVIEWERS' COMMENTS

Reviewer #1 (Remarks to the Author):

The authors have appropriately responded to my concerns. I consider this highly appropriate for publication in Nature Communications.

Reviewer #2 (Remarks to the Author):

The authors have complied with all the reviewers comments and I am very happy to recommend that this article be published.

Reviewer #3 (Remarks to the Author):

I would like to thank the authors who have addressed my comments point-by-point. I was convinced by their response. Now I have no further comment to the manuscript. But it seems that there are a few technical mistakes in the supplementary material. For example, the tables in S3 should be not Tables S1 and S2 but Tables S5 and S6, respectively, and the authors must cite the correct table number in the text.

We once again thank all the reviewers and the editorial staff at Nature Communications for their valuable input that helped improve the manuscript.

Reviewer 3:

1. I would like to thank the authors who have addressed my comments point-by-point. I was convinced by their response. Now I have no further comment to the manuscript. But it seems that there are a few technical mistakes in the supplementary material. For example, the tables in S3 should be not Tables S1 and S2 but Tables S5 and S6, respectively, and the authors must cite the correct table number in the text.

Author Comment:

Thank you for pointing this out. The table numbers have now been changed to adhere to Nature Communications guidelines, and the error in the chronological order has been corrected.